# Sicilian Rivet Wheat Landraces: Grain Characteristics and Technological Quality of Flour and Bread

**DOI:** 10.3390/plants12142641

**Published:** 2023-07-14

**Authors:** Alfio Spina, Paolo Guarnaccia, Michele Canale, Rosalia Sanfilippo, Michele Bizzini, Sebastiano Blangiforti, Silvia Zingale, Angela Roberta Lo Piero, Maria Allegra, Angelo Sicilia, Carmelo Nicotra, Umberto Anastasi

**Affiliations:** 1Council for Agricultural Research and Economics (CREA), Research Centre for Cereal and Industrial Crops, Corso Savoia, 190, 95024 Acireale, Italy; michele.canale@crea.gov.it (M.C.); rosalia.sanfilippo@crea.gov.it (R.S.); 2Department of Agriculture, Food and Environment, University of Catania, Via Santa Sofia 98, 95123 Catania, Italy; silvia.zingale@phd.unict.it (S.Z.); rlopiero@unict.it (A.R.L.P.); angelo.sicilia@unict.it (A.S.); umberto.anastasi@unict.it (U.A.); 3Stazione Consorziale Sperimentale di Granicoltura per la Sicilia, 95041 Caltagirone, Italy; michele.bizzini@gmail.com (M.B.); blangiforti@granicoltura.it (S.B.); carmelo.nicotra@granicoltura.it (C.N.); 4Research Centre for Olive, Fruit and Citrus Crops, Council for Agricultural Research and Economics (CREA), Corso Savoia, 190, 95024 Acireale, Italy; maria.allegra@crea.gov.it

**Keywords:** bread, dough, cluster analysis, grain characteristics, rheological indices, rivet, tetraploid wheat, *Triticum turgidum* subsp. *turgidum*

## Abstract

In recent years, the growth of tetraploid Sicilian wheat landraces has been arousing increasing interest. In this study, eighteen local genotypes of *Triticum turgidum* subsp. *turgidum*, belonging to the groups ‘Bufala’, ‘Ciciredda’, ‘Bivona’ and ‘Paola’, and two cultivars of *Triticum turgidum* subsp. *durum* (the old variety ‘Bidì’, and a more recent variety ‘Simeto’) were assessed for the characteristics of the grain and bread-making performance of their flours and doughs, as well as the quality of the loaves. The grain of the twenty genotypes came from a field trial conducted during 2018–2019 in south-eastern Sicily. The main commercial features of the grain (thousand kernel weight and hectolitre weight), including the defects (starchy, black pointed and shrunken kernels), were determined. The wholemeal flours and doughs obtained from the grain of each genotype were evaluated for the main technological quality (physico-chemical and rheological characteristics), and processed into loaves, whose main quality indices (volume, height, weight, moisture and porosity) were assessed. The results from such analyses allowed the authors to evaluate the genotypes’ bread-making suitability. In particular, for the grain characteristics, hectolitre weight varied from 68.23 (‘Bufala Rossa Lunga 01’) to 77.43 (‘Bidì 03’) kg/hL, passing through the typical values for common and durum wheat. Among the grain defects, the black point defect was absent in all the grain samples, except for that of ‘Bufala Nera Corta 01’ (2%). Dry gluten content varied from 6.22 to 10.23 g/100 g, and sedimentation test values were low or medium-low, with values ranging from 22 to 35 mL. Amylase activity was low and highly variable among the genotypes, with the maximum value observed for ‘Bufala Rossa Corta b01’ (509 s). The doughs evidenced a poor quality for bread making with alveograph values of W ranging from 12 to 145 (10^−4^ × Joule) and thus the volume of the loaves varied from 346.25 cm^3^ of ‘Bivona’ and ‘Ciciredda’ to 415.00 cm^3^ of ‘Bufala Rossa Lunga’. A Tandem Cluster Analysis was conducted on a set of all the response variables. The Hierarchical Cluster Analysis was initially run. A five-cluster solution identified three clusters further segmented and two single branches. Overall, the study highlighted the possibility of using some of these landraces alone for the production of traditional breads locally appreciated or together with other ingredients for the production of crumbly baked goods such as substitutes for bread and biscuits.

## 1. Introduction

Wheat is one of the primary sources of nutrition, and is thus a key crop for meeting the food demand of the ever-growing global population [1]. Accordingly, thanks to its great adaptability to different environments, and growing conditions, wheat is one of the most widespread cereals along with maize and rice, contributing some 20% of the total dietary calories for an estimated global population of about 10 billion people in 2050 [2], and provide various essential ingredients for the human and animal diet [3].

Italy plays a considerable role in durum wheat production in Europe, mostly as a result of the economic value of the pasta industry, which has pushed the intense breeding work conducted since the beginning of the 20th century [4]. Sicily, in particular, where durum wheat semolina is also used for the production of bread, is the second largest Italian region in terms of production of this cereal, with just under 264 × 10^3^ ha cultivated and slightly more than 682 × 10^3^ tons of grain produced, according to ISTAT (2022) [5].

Despite the thriving activity of Italian industries connected to cereal production, the sector has been going through a severe crisis for several years due to the lack of self-procurement of raw materials, durum and soft wheat in particular, to be destined for excellent products such as pasta and bread. Furthermore, the cereal sector can sometimes contribute significantly to the waste of energy, land and water resources and thus to climate change, if the cropping and processing phases are not carefully managed [6]. Likewise, the productivity and quality of durum wheat, and therefore the profitability for producers, are compromised by climate uncertainty, primarily due to drought and heat stress experienced by the crop. On the other hand, there is also a lack of varieties suitable for low-input farming systems, particularly organic ones. Indeed, the improved semi-dwarf varieties that have gradually replaced the landraces require many auxiliary cultivation inputs [7,8,9,10]. Conversely, landraces and old varieties are characterised by greater plant height and wide genetic variability, making them more suitable to the pedoclimatic conditions under which they have been traditionally cultivated. Therefore, such germplasm represents an important source of useful traits [11,12,13,14], which should be preserved, explored and valorised, especially in marginal areas, where crop diversification plays a key role in achieving environmental and economic sustainability. New breeding projects are needed to redefine a new ideotype of the wheat plant that is suitable for low-input farming, resilient to the effects of climate change and possesses an increased nutritional value [15].

Recently, a molecular genotyping approach combined with the evaluation of a set of morpho-agronomic and productive traits was used to evaluate a group of tetraploid wheat landraces (rivet wheat) called ‘*Bufala*’ [16]. These genotypes are traditionally cultivated in the mountain areas of Sicily, and are characterised by high adaptability in terms of cold tolerance, ability to grow in marginal soils, competitiveness against weeds and tolerance/resistance to some biotic stress [16]. ‘*Bufala*’ genotypes are mostly grown in Caronie (Cesarò, Bronte and Randazzo) and Peloritani (Novara di Sicilia) in Sicilian mountains. Such genotypes display different morphological traits, such as glume colour and pubescence, and brown colour. Some of these rivet landraces are classified as ‘*Bufala lunga*’ due to their very long ears, and are very similar to ‘*Paola*’ [17]. An interesting option of promoting the cultivation of such wheats landraces is their introduction in organic cropping systems, which do not involve the use of mineral fertilisers and herbicides. Moreover, landraces and old varieties have recently gained increasing interest among consumers, who tend to identify the derived products as more genuine and healthier, also because these products are often prepared from wholemeal and therefore contribute to the adequate intake of micronutrients and bioactive compounds [18]. Regarding the latter, indeed, some studies also found significant differences between landraces and improved varieties of wheat in terms of carotenoids, dietary fibre, polyphenols, and flavonoids [19,20].

These ‘ancient grains’ may represent a useful compromise, combining sufficient productivity, quality and sustainability, especially when used in local and speciality supply chains. In the context of the latter, such genotypes can be valorised through the rediscovery of traditional recipes and customs, and the development of new products, thus increasing genetic diversity in farming systems and food diversity in diet regimes. Accordingly, although these genotypes will hardly reach the production level of the modern varieties due to their lower yield ability and unsuitability to high-input agronomic management, they can satisfy consumers’ demand for local products with high quality and typical taste [21].

For the above reasons, a comparative experiment including eighteen *T. turgidum* subsp. *turgidum* landraces (rivet wheat) and two *T. turgidum* subsp. *durum* (durum wheat) genotypes was conducted to evaluate the grains’ main characteristics, the bread-making suitability of the wholemeal flours and doughs, and the quality of the derived loaves.

## 2. Results and Discussion

### 2.1. Grain Characteristics, Physico-Chemical and Rheological Features and Bread-Making Quality

The grain characteristics of the studied genotypes are reported in Table 1. A huge variability was observed for all the features examined, especially for the grain defects (e.g., starchy, shrunken, and black pointed kernels), which reflects the effect of the interaction between the genotype and environmental conditions, mainly drought and high temperature, during the grain filling phase of the crop.

Hectolitre weight was equal to 73 kg/hL, on average, and was significantly different among the tested cultivars, ranging from a minimum in ‘Bufala Rossa Lunga 01’ to a maximum in ‘Bidì 03’. The observed values intercept the range typical for common (72–74 kg/hL) and durum wheat (78–80 kg/hL), due to the genetic constitution of these rivet wheat landraces (*Triticum tugidum* subsp. *Turgidum*), whose kernel has a mealier consistency. Hectolitre weight values reported by other authors on the modern and ancient varieties of durum wheat are in line for Simeto [22,23] and ‘Bidì’ [23], while they are slightly higher for ‘Paola’. Thousand kernel weight (TKW), recognised as very important for grain quality [24], was significantly greater for the durum wheat cultivars, mainly for ‘Simeto’, but also for ‘Bidì 03’. Compared to the latter, however, a group of ‘Bufala’ landraces evidenced similar values of TKW (‘Bufala Nera Lunga 01’, ‘Bufala Nera Lunga 02’, ‘Bufala Nera Corta 02’, ‘Bufala Nera Lunga 04’, ‘Bufale Salice 01’). Values found for Bidì and Simeto, 58.67 g and 66.63 g, respectively, were higher than those found by other authors [23,25,26], while with respect to ‘Paola’, the values were lower than those indicated by Guarnaccia et al. (2020) [23].

Among the defects of the grain, the percentage of starchy and shrunken kernels varied greatly in the tested genotypes, while the black point defect was absent in all the grain samples, except for that of ‘Bufala Nera Corta 01’, which had a low value. Accordingly, the black point defect lies in a grain discolouration that degrades the quality of the grain, with negative consequences in terms of commercial value [27,28]. It is a defect strongly conditioned by abiotic and biotic factors, such as heavy rains, high humidity, and extreme temperatures [29], favouring fungal development, especially *Alternaria alternata* [30]. Its incidence may depend on either severity of the stresses or the response of the genotypes [31].

Starchy kernels consist of whiteness or lack of translucency [32], and are mainly caused by environmental stresses during the growing season. Besides the defect, starchy kernels are also associated with a reduction in protein content [33,34]. Within the ‘Bufala’ group, the starchy kernels percentage was lower for ‘Bufala Rossa Lunga 01’ and for other landraces with similar values. However, the lowest percentage was found in the control durum wheat variety ‘Simeto’.

Shrunken kernels are stunted kernels produced due to an anomalous time course of grain filling and maturation due to heat, drought, or biotic stresses [34,35]. Additionally, for the shrunken kernels defect, the results of this study highlighted some variability, with certain genotypes showing low or no incidence (‘Bidì 03’), with others being more affected, such as ‘Bufala Nera Corta 01’ and ‘Bufala Nera Corta 02’, which presented the highest values, but were not statistically different compared to those of other ‘Bufala’ and ‘Bufala’-related landraces (‘Ciciredda’, ‘Bufala Rossa Corta b01’, ‘Bufala Rossa Lunga 01’).

In line with Venora et al. [34], these results stressed the need to carefully evaluate the intrinsic and extrinsic factors responsible for starchy and shrunken kernel incidence to minimise the associated negative effects on grain and semolina yield and quality. Even with optimal protein content, the presence of these defects causes a deterioration in the overall quality characteristics of the flour, and should not be overlooked.

As observed by other authors [36,37] on durum wheat, environmental aspects significantly impact the physical and chemical characteristics of seeds and semolina, causing a certain variability even among the same cultivars grown in different soil and climatic conditions.

Moisture (Table 2), measured and expressed on a dry matter basis, showed high values, which did not differ among the studied genotypes, and were within the range indicated by Italian law (DPR 187/2001) [38], which sets the maximum limits for moisture in durum wheat semolina (14.50–15.50%).

The protein content, expressed on a dry matter basis (Table 2), was highly variable, ranging overall from just under 10 to about 15 g/100 g d.m., and which differed significantly among the different genotypes. These results are in line with those found by other authors for different genotypes of ‘Bufala’ (Nera, Rossa and Bianca) [39], ‘Simeto’ [40] and ‘Bidì’ [25]. Two-thirds of the tested genotypes evidenced a protein content higher than the minimum value established by the DPR 187/2001 [36], i.e., 10.50%. On the other hand, for the landraces ‘Bivona 03’, ‘Bufala Bianca 03’, ‘Bufala Nera Lunga 04’, ‘Bufala Cerami 01’, ‘Bufala Troina 01’ and ‘Ciciredda 03’, values below this legal limit were observed. The values found for the two control durum wheat genotypes ‘Simeto’ and ‘Bidì 03’, instead, were in a different range from those reported by other studies [41,42].

Almost all the genotypes tested, except for ‘Bivona 03’, ‘Bivona 04’, ‘Bufala Bianca 03’, and ‘Bufala Nera Corta 02’, reported ash content slightly higher than the limits specified by Italian law 0.90% [38]. The gluten analysis was carried out using the glutomatic apparatus, except for the samples ‘Bufala Nera Lunga 04’, ‘Bufala Cerami 01’, ‘Bufala Salice 01’ and ‘Bufala Troina 01’, in which the gluten was manually extracted because it was particularly weak. The values of both wet and dry gluten (the latter should preferably be used, according to the Italian law 580/1967 [43]) varied significantly among the genotypes, although to a different extent. The values of gluten index varied considerably, with appreciable differences among the studied genotypes, and with the lowest value being observed for ‘Bufala Cerami 01’, compared to modern varieties such as Simeto, with values higher than 90%, as also reported by other authors [25,44]. Almost half of the genotypes reveal weak or very weak gluten levels, whereas the others have a gluten ranging from less strong to moderately strong. In particular, among the landraces, ‘Bivona 03’, ‘Bivona 04’, ‘Bufala Bianca 02’ and ‘Bufala Bianca 03’ exhibited significantly higher values of gluten index, similar to the control genotypes, namely ‘Simeto’ and ‘Bidì 03’. In line with other authors [45,46,47], these data confirm the weaker gluten content of the Sicilian tetraploid wheat landraces compared to the improved varieties [48]. A large variability within the studied set of cultivars was also observed with respect to water binding capacity. Among the landraces, the minimum and maximum absorption values were found for ‘Bufala Cerami 01’ and ‘Bufala Nera Lunga 02’, respectively, although other genotypes evidenced values that were not statistically different.

These results are coherent with those reported by Fiore et al. [49] on rivet wheats.

Data on the colour of the semolina are reported in Table 3, from which the appreciable variability among the assessed genotypes can be observed. Specifically, the brown index of the rivet landraces varied from the lowest value for ‘Ciciredda 03’ to the greatest value for ‘Bufala Corta Rossa b01’, which was slightly but significantly different from that of the control durum wheat variety ‘Simeto’. By contrast, the red index (*a**) showed a narrow range among the genotypes, with significantly lower values observed for both ‘Bufala Rossa Lunga 01’ and ‘Ciciredda 02’. A significantly lower yellow index (*b**) was found for ‘Ciciredda 03’. Among the landraces, only ‘Bufala Rossa Corta b01’ had a higher yellow index value, being significantly but slightly lower than those of the control varieties ‘Simeto’ and ‘Bidì’, which had values in agreement with other authors [25,40]. The higher yellow index (*b**) found for ‘Simeto’ compared to those of the Sicilian landraces demonstrates the effect of genetic selection on this important commercial and health-related trait [49], due to the higher content of carotenoids, which are responsible for both the yellow colour of the semolina and protective effects against free radicals.

The SDS sedimentation test allowed us to evaluate the qualitative and quantitative aspects of the semolina based on the protein content, particularly on the characteristics of gluten, as well as the suitability of the semolina for bread making. The values of this feature were quite variable, from a minimum found for ‘Bufala Troina 01’ to a maximum observed for ‘Bufala Rossa Lunga 01’ (33.00 mm) and ‘Bufala Nera Corta 01’ (31.50 mm), which did not differ from those of the two control varieties of durum wheat ‘Bidì 03’ (32.00 mm) and ‘Simeto’ (35.50 mm), which present slightly lower values than those found by other authors who indicate a range between 43 and 45 mm for both [23,44]. Lower SDS values were also measured for ‘Paola’, compared to what was reported by other authors [23]. These results highlighted the minor sedimentation heights, which indicate the low quality of gluten of the semolina.

The falling number analysis showed a low or medium amylase activity, as demonstrated by the range of values reported in Table 4.

Despite the low amylase activity, most of the studied cultivars’ gluten quality, measured via the SDS test, showed average values. Due to the lack of specific references on the amylase activity and sedimentation test for the considered genotypes, it was useful to compare the results obtained within this study with those reported for durum (*Triticum turgidum* subsp. *durum*) and soft (*Triticum aestivum* subsp. *aestivum*) wheat genotypes finding lower values, as also reported by other authors [50,51,52,53]. From the data obtained by the mixograph, it is possible to note a high gap between the values of mixing time, equal to 163 s between ‘Bidì 03’ and ‘Bufala Salice 01’. In the case of the peak dough height, the values were around 3 (M.U.), with higher points recorded for both control durum wheat genotypes ‘Bidì 03’ and ‘Simeto’, as well as for ‘Bufala Rossa Lunga 03’.

Moreover, from alveograph analysis (Table 5) it was possible to point out that the doughs obtained with the ‘Bufala’ semolina were very weak, and in some cases very extensible, with P/L lower than 1. In particular, the alveographic W values of rivet wheat landraces were lower than 50 × 10^−4^ × J, thus indicating excessive weakness, except for ‘Bufala Rossa Corta b01’, which exhibited a similar value to the control genotype of durum wheat ‘Bidì 03’ with P/L and W values slightly lower than those obtained by other authors [25], but still lower than that observed for ‘Simeto’, whose values were in agreement with what was found by other authors for P/L [25,44] and slightly lower for W [44].

As far as the P/L ratio is concerned, the obtained results suggested heterogeneous dough behaviours, as demonstrated by the wide range of values, equal to 1.24, between ‘Bufala Bianca 03’ and ‘Bufala Rossa Lunga 01’. In contrast, a group of genotypes, including the ones with values from 0.85, such as ‘Bufala Salice 01’, and 1.67, such as ‘Bufala Rossa Lunga 01’, without considering the control durum wheat varieties, was regarded as semolinas giving rigid, and not extensible, doughs.

Given this, the W values along with the P/L ratios indicated the marked and different suitability of these semolinas for the production of bakery products, especially biscuits and similar, while at the same time suggesting the difficulty of using these as such.

As far as the farinograph analysis was concerned, the results showed significant differences. The water absorption was also variable among the tested genotypes. It was equal to 51%, on average, ranging from the minimum value observed for ‘Bufala Nera Lunga 04’ and the maximum found for ‘Simeto’, with values equal to 63% that confirm what was reported by Raffo et al. (2003) [40]. This poor absorption capacity is mainly attributable to the low protein content and low gluten index, which actively condition the rheological characteristics. This effect is also evident in the other indices measured using the farinograph, i.e., development time and dough stability. The development of the dough is higher in the two control durum wheats ‘Simeto’, which is in agreement with the results of other authors [40], and ‘Bidì 03’, which have greater values for protein and gluten, and tend to decrease for the other genotypes with minimum values being observed in ‘Paola 02’ and ‘Bufala Nera Corta 02’. Conversely, the stability of the dough also had the same behaviour, highlighting higher values in ‘Simeto’, as indicated by other authors [40], and lower in ‘Bufala Nera Corta 01’. As expected, the degree of softening was higher for the genotypes with reduced development times and stability of the dough, ranging between a minimum value for ‘Simeto’ and a maximum value for ‘’Bufala Nera Corta 01’. The alveograph and farinograph data show higher values in the modern cultivars than in the ancient ones, confirming the gluten and SDS values obtained, as observed by other authors [54].

Ultimately, samples of modern varieties such as ‘Simeto’ were harder and more resistant [55] than the rivet samples studied.

By analysing the results reported in Table 6, related to the physical characteristics of the bread loaves (volume, height, weight, moisture and porosity), it was possible to observe the variable performances of the tested genotypes, with differences being statistically appreciable among the mean values. Precisely, the higher volume value among the rivet landraces was observed for ‘Bufala Rossa Lunga 03’, ‘Bufala Rossa Corta b01’ and ‘Bufala Nera Lunga 02’, which showed bread-making volumes above 400 cm^3^, thus being similar to that found for the durum wheat control variety ‘Bidì 03’.

The latter also evidenced a greater height, together with ‘Bivona 04’, ‘Bufala Rossa Corta b01’ and ‘Bufala Rossa Lunga 03’. The weight was higher for ‘Simeto’, and lower for ‘Bufala Bianca 03’. These three indices provide an indication of more compact loaves, which tend not to be suitable for long leavening, because of the low protein and gluten quality. Accordingly, this gluten network is not really capable of optimally retaining the CO_2_ and ethyl alcohol produced during the leavening phas, and providing support to the dough to allow its volumetric growth. Furthermore, the low amylase activity reported in Table 4 indicates that yeasts will have a lower amount of simple sugars deriving from enzymatic action, which also conditions their growth and development. Accordingly, with the exception of ‘Bufala Bianca 03’, ‘Bidì 03’, ‘Bufala Nera Lunga 01’ and ‘Bufala Troina 01’, the values of porosity indicated the more compact crumbs of loaves with low development of pores (values 6 and 7).

Finally, the maximum moisture value was found for ‘Bufala Nera Corta 01’, while the lowest was found for ‘Bufala Bianca 02’, highlighting the higher variability among the loaves, because of the different water retention capacity during cooking. Of course, the increase in moisture also affected the final weigh of loaves.

Furthermore, Table 7 shows the colorimetric data of the bread, both for the crumb and the crust. For the luminosity of the crumb, we did not find any statistical differences among the tested genotypes, with the lowest values being obtained for ’Bufala Bianca 03’ and the highest ones for ‘Bivona 03’, whereas in the crust, a greater variability was observed, with a range equal to 10.13 between ‘Bufala Bianca 02’ and ‘Bufala Troina 01’ and some significant differences among the studied genotypes. The red index values of the crumb varied from a minimum value of ‘Bufala Troina 01’ to a maximum value of ‘Bufala Nera Corta 02’, while in the crust, closer values between ‘Bufala Nera Lunga 02’ and ‘Bufala Nera Lunga 04’ were observed. The yellow index in the crumb follows the same trend evidenced by the red index, with a maximum value for ‘Simeto’ and a minimum one for ‘Bivona 04’.

In the crust, a wide variability was recorded, as well as in the crumb, with a range equal to 9.86 between the lowest for ‘Bufala Rossa Corta b01’ and the highest for ‘Bufale Salice 01’.

The colorimetric indices of the loaves indicated differences among the genotypes, which reflected what was previously found for the semolina, as a consequence of the genotypic and environmental effects. These aspects were more evident in the crumb than in the crust, since in the latter the cooking effect, with the consequent Maillard reaction, tends to make the bread more uniform.

### 2.2. Cluster Analysis (Hierarchical Cluster Analysis and K-Means Cluster Analysis)

A tandem of two cluster analysis techniques (Hierarchical Cluster Analysis and K-means Cluster Analysis) was conducted on a set of all the response variables. The Hierarchical Cluster Analysis was initially run to identify useful patterns, and consequently the number of clusters within the large data set, with no a priori information. As variables were measured on different scales, to neutralise the impact of variables with large values on distance measurements with respect to variables with small values, the Hierarchical Clustering procedure involved calculating standardised scores for the variables. The standardised scores were saved as new Zscore-variables. The data treatment resulted in all variables contributing more equally to the distance measurement based on the squared Euclidean distance and the Ward Linkage method of clustering. Figure 1 shows the dendrogram resulting from Hierarchical Cluster Analysis. A five-cluster solution identified three clusters further segmented and two single branches (‘Bidì 03’ and ‘Simeto’, the control varieties). The first cluster consisted of ‘Bufala Nera Lunga 01’, ‘Bufala Nera Lunga 02’, ‘Bufala Rossa Corta b01’ and ‘Bufala Rossa Lunga 03’; the second cluster included ‘Bivona 03’ and Bivona 04’, ‘Bufala Bianca 02’, ‘Bufale Troina 01’, ‘Ciciredda 03’, ‘Paola 02’, ‘Bufale Cerami 01’, and ‘Bufala Nera Lunga 04’; the last cluster included ‘Bufala Nera Corta 01’ and ‘Bufala Nera Corta 02’, ‘Bufale Salice 01’, ‘Bufala Rossa Lunga 01’, ‘Ciciredda 02’, and ‘Bufala Bianca 03’.

This five-cluster solution was the conclusion of the Hierarchical Cluster Analysis that was used as input to the K-means Cluster Analysis algorithm to assign and interpret the cluster membership of rivet and control varieties.

The cluster membership obtained from the K-means Cluster Analysis is reported in Figure 1. The first cluster included ‘Bidì 03’ and ‘Bufala Rossa Corta b01’; the second only ‘Simeto’; the third cluster included ‘Bufala Nera Corta 01’, ‘Bufala Rossa Lunga 01’ and ‘Ciciredda 02’; the fifth was a large cluster including ‘Bivona 03’ and ‘Bivona 04’, ‘Bufala Bianca 02’ and ‘Bufala Bianca 03’, ‘Bufala Nera Corta 02’, ‘Bufala Nera Lunga 04’, ‘Bufale Cerami 01’, ‘Bufale Salice 01’, ‘Bufale Troina 01’, ‘Ciciredda 03’ and ‘Paola 02’.

Considering the distance to the cluster centre (Appendix A), the smaller the value, the more representative that rivet or control variety was of that cluster, the closer it was to what is called the centroid, e.g., the middle of that cluster. Conversely, the rivet or tester with the highest value was the least representative of that cluster. From our results, the distances of the two components of cluster 1 were quite similar; ‘Ciciredda 02’ was the centroid of cluster 3, ‘Bufala Nera Lunga 02’ of cluster 4, and ‘Paola 02’ of cluster 5.

Based on the data presented in the ANOVA table (Appendix A), the significance level cannot be used to test the hypothesis regarding the mean variables, as the results from the dispersion analysis are purely descriptive, since the groups are formed deliberately by the distance between them in the multidimensional space. However, examining the differences between the F-ratios made it possible to draw meaningful conclusions about the role of different mean variables in cluster formation. The variables P/L, W, and semolina_*b** had the maximum influence in forming the clusters. On the other hand, the variables Crumb_*L**, Black point, and Crumb_Red index had the lowest effect.

Figure 2 showed the qualitative footprint of each cluster. Cluster 2, which was represented solely by ‘Simeto’, stood out as having the highest value for most variables. In contrast, the members of cluster 5 exhibited the lowest average values.

## 3. Materials and Methods

### 3.1. Materials

A total of twenty tetraploid wheat genotypes, including eighteen Sicilian rivet wheat (*Triticum turgidum* L. subsp. *turgidum*) landraces belonging to ‘Bufala’ and ‘Bufala’-related genotypes (genotypes genetically close to the ‘Bufala’ germplasm) and two improved varieties of durum wheat (*Triticum turgidum* L. subsp. *durum*), specifically an old variety (‘Bidì 03’) and a more recent one widespread in Sicily (‘Simeto’), used as control genotypes, were chosen for this study. The grains were provided by the ‘Stazione Consorziale Sperimentale di Granicoltura per la Sicilia’ (Santo Pietro, Caltagirone, Catania, Italy), and came from an agronomic field trial laid out according to a randomised blocks design with three replicates, conducted during the 2018–2019 growing season at the experimental station located in ‘Vaccarizzo’ (Lat. 37,119,000–Lon. 14,521,000°–316 m a.s.l.) (Santo Pietro, Caltagirone, Catania), adopting a low input agronomic management consisting of 30 kg ha^−^^1^ of N supplied at sowing and without application of other chemical inputs to control weeds, pests and diseases during the cropping season.

### 3.2. Determination of Grain Characteristics and Grain Milling

Representative samples of grain per plot were used to determine the thousand kernels weight (TKW), test weight (TW), starchy kernels, shrunken kernels and black-pointed kernels. TKW was obtained by weighting 8 sub-samples of 100 kernels and the average weight was related to 1000 kernels. TW was determined with a Test Weight Module (TWM) installed under the Infratec 1241 Grain Analyser (Foss Tecator, Höganas, Sweden). The starchy kernels, shrunken kernels and black-pointed kernels percentages were visually estimated on representative sub-samples of kernels (30 g). Kernels were sorted visually into wholly vitreous grains and not (at least two spots), and the latter was expressed as a percentage of the total kernels. Worldwide recognised procedures define fully vitreous kernels as ‘those that do not disclose the least trace of farinaceous endosperm’ [56].

The grain from the three field replicates was milled using an experimental mill to obtain semolina (Bona, Monza, Italy).

All analyses were performed in triplicate.

### 3.3. Physico-Chemical Analyses of Semolina

Moisture and protein content (% dry matter) was determined by means of Infratec 1241 Grain Analyser (Foss Tecator, Hoganas, Sweden) by near-infrared transmittance using, for protein content, a calibration (range 8.3 to 15.3) based on the Kjeldahl nitrogen method. The gluten quantity (wet and dry gluten content) and quality (gluten index) were obtained using a Glutomatic 2200 apparatus, a Centrifuge 2015 and a Glutork 2020 (Perten Instruments AB, Huddinge, Sweden) according to the ICC Standard No. 158 [57] and to the AACC method 38-12.02 [58], respectively. Centrifugation was performed to force the wet gluten through a specially constructed sieve under standardised conditions. Wet gluten passing through the grid is called the “B fraction”, and when this is highly represented, it indicates the poor technological quality of the gluten.

The semolina was evaluated for its moisture content according to the AOAC 935.25 method [59], by drying in a Memmert oven at a temperature of 103 °C, up to constant weight. The results were expressed as relative percentage moisture (RH%) [60].

The CR 200 Minolta Chroma colourimeter (Minolta, Osaka, Japan) was used to evaluate the colour in semolina and bread loaves. The CIELab colorimetric model was adopted [61] to express the results through the following indices: *L** [61,62,63], *a** (red index), and *b** (yellow index) [64,65].

All the analyses were performed in triplicate.

### 3.4. Technological Tests on Doughs of Semolina

Mixographic analyses were performed to measure the strength of the dough even from small quantities of semolina. The mixography curve was obtained using the National Mfg. Co. (Lincoln (NE), USA), following the method AACC 54-40.02 [66].

The α-amylase activity was determined using the Falling Number 1500 apparatus (Perten Instruments AB, Huddinge, Sweden), following the method described in ISO 3093 [67]. The sodium dodecyl sulfate (SDS) sedimentation test (Sigma-Aldrich, Milan, Italy) is a useful preliminary test for estimating gluten quality. This test was performed according to the method of Dick and Quick [68]. The farinograph measures the mechanical resistance of the dough during kneading as it is characterised by three specific moments. The first is characterised by the absorption of the added water and the formation of the gluten mesh; the second includes a stability phase in which the disulphide bonds are continuously broken and reformed; in the third, the gluten mesh breaks, and the curve descends [69].

The farinographic curves were obtained using the 300 g mixing chamber using the Brabender farinograph (Duisburg, Germany), according to the method AACC 54–21.02 [70].

The alveographic test was conducted using an alveograph (Tripette et Renaud, Villeneuve-la-Garenne, France), according to the standard method [71], equipped with Alveolink ng software V1.04/99.

All the analyses were performed in triplicate.

### 3.5. Baking Test

For the baking test, loaves were obtained according to the official methodology AACC 10-10.03 [72], as described by Ficco et al. [73]. Their physical characteristics, such as volume, height, weight, moisture, crumb porosity and the colour of the crumb and crust were evaluated. Specifically, the volume was determined based on the displacement of rape seed in a loaf volume meter according to method AACC 10-05 [74]; loaf height was measured using a digital calliper (Digi-MaxTM, Scienceware^®^, NJ, USA). Weight was measured using a digital scale (OHAUS mod. Adventurer pro AV2102C, Pine Brook, NJ, USA).

For the assessment of crumb porosity, the central bread slices of each loaf were taken and visually compared with the eight Dallmann reference images, which represent the cross-section of loaves with different crumb structures. The porosity of the crumb was evaluated on the basis of the 8-degree Mohs scale as modified by Dallmann [75]. According to this scale, 1 indicates a non-uniform structure, with large and irregular cells, and 8 indicates a compact uniform structure, with small and regular cells [76].

The moisture and the colour of both the crust and crumb of the loaves were assessed according to the method cited above for semolina.

All the measurements were performed in duplicate.

### 3.6. Statistical Analyses

The data of each response variable were subjected to a one-way ANOVA after the evaluation for homogeneity of variances based on Bartlett’s test. Tukey’s test was then applied to establish whether there were significant differences between the means of the genotypes (*p* ≤ 0.001). Percentage data of starchy kernels, shrunken kernels and black-pointed kernels were arksin transformed before the application of the ANOVA procedure. The results are reported as means ± standard deviation.

Hierarchical Cluster analysis was applied, as an unsupervised data analysis technique to explore the naturally occurring groups within a dataset. Its goal was to classify cases into groups that are relatively homogeneous within themselves and heterogeneous with respect to each other, based on a defined set of response variables, which in this case includes all the studied variables.

The squared Euclidean distance, which is the sum of the squared differences over all of the variables, was used. Distance is a measure of how far apart two objects are, while similarity measures how similar two objects are. For cases that are alike, distance measures are small and similarity measures are large.

Hierarchical cluster analysis was conducted to initially determine the number of clusters in our data, and then its result was used as an input to the K-means algorithm, a non-hierarchical cluster analysis belonging to the same class of techniques of the Hierarchical Cluster analysis.

Utilising a tandem of effective methodologies, such as the Hierarchical Cluster analysis for determining optimal cluster numbers and the K-means for executing clustering and extracting meaningful insights, provided a powerful combination of these tools. This combination allowed us to determine the similarities and dissimilarities among groups of selections, highlighting their distinctive characteristics in comparison to other groups.

The statistical analysis was conducted with the IBM SPSS Statistics software package version 20 (IBM Corporation, Armonk, NY, USA, 2011).

## 4. Conclusions

In this study, the quality of grain, semolina, and bread of tetraploid rivet wheat landraces grown in Sicily was assessed. The focus was on the ‘Bufala’ and ‘Bufala’-related genotypes, which represent an important source of genetic diversity, particularly suited to low-input farming systems. With the main aim of filling the gap in the specialised literature about these landraces’ merceological, physical, chemical, and technological quality aspects, the width of the quality variations due to the genotype factor was defined, with particular emphasis on bread-making suitability. Overall, the data obtained evidenced non-optimal physical, chemical, and technological characteristics of the flour and dough for bread making. In particular, the results of both the gluten quality analysis and the technological indices (e.g., mixograph, farinograph and alveograph analyses) highlighted that the flours of the genotypes under study were weak, with reduced strength and dough stability, and were structurally unstable. Only the pure flours of ‘Bufala Rossa Lunga 03’, ‘Bufala Rossa Corta b01’, and ‘Bufala Nera Lunga 02’ were suitable for bread making. All the other landraces could be used in bread making when mixed with other genotypes of durum and bread wheat flour. Considering this, their use in mixtures with other flours could be suggested in the preparation of crumbly bakery products, such as bread substitutes and biscuits. In addition, it was proposed to use boiled whole grain to prepare the so-called ‘cuccìa’, which, being both sweet and savoury, is a widely appreciated traditional Sicilian product. Furthermore, the grain of these genotypes can be used to produce typical mountain bread and to prepare excellent soups. In the Hierarchical Cluster Analysis, a five-cluster solution identifies three clusters further segmented and two single branches. This five-cluster solution is the conclusion of the Hierarchical Cluster Analysis that was used as input to the K-means Cluster Analysis algorithm to assign and interpret the rivet and tester cluster membership. The variables P/L, W, and semolina_*b**, had the maximum influence on the formation of the clusters. On the other hand, the variables crumb_*L**, black point and crumb_red index had the least influence.

In this sense, food companies have long been moving towards the production and marketing of functional and sustainable wheat-based snacks through the inclusion in the formulation of healthy ingredients. In line with this, it would be desirable to promote the research on the development of formulations, including these semolinas and other ingredients, acting simultaneously as structuring and health components, supporting small-scale farming systems based on low-input agronomic management, particularly organic farming, while also promoting a direct producer–consumer relationship. 

## Figures and Tables

**Figure 1 plants-12-02641-f001:**
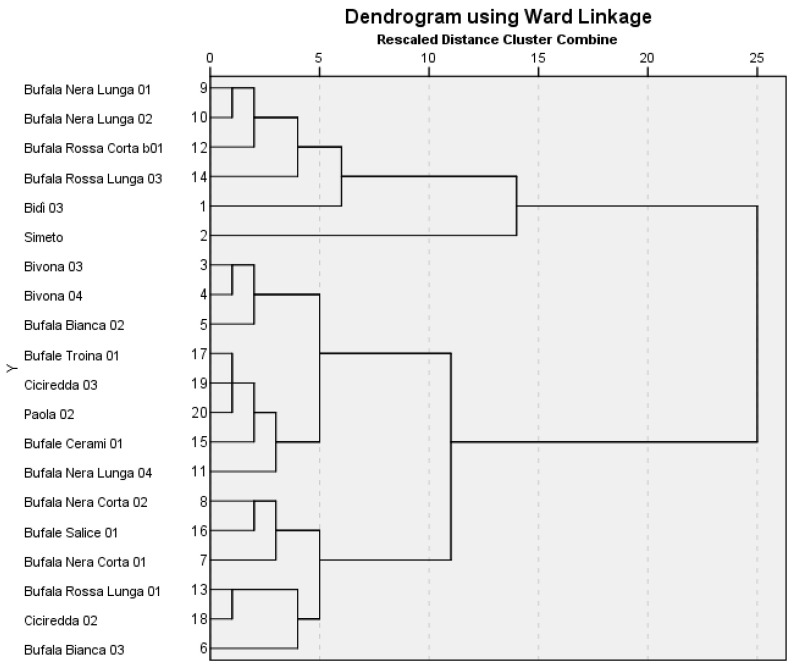
Dendrogram of Hierarchical Cluster Analysis.

**Figure 2 plants-12-02641-f002:**
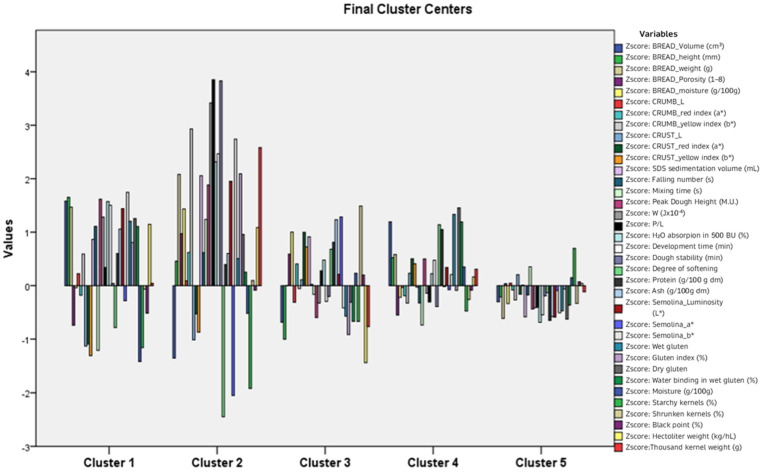
K-means Cluster Analysis. Qualitative footprint.

**Table 1 plants-12-02641-t001:** Physical characteristics of kernel in the Sicilian landraces (means ± standard deviations). For all the response variables, different letters indicate a significant difference between the means (*p* ≤ 0.001).

Sample	Hectolitre Weight (kg/hL)	Thousand Kernel Weight (g)	Starchy Kernels (%)	Black Pointed Kernels (%)	Shrunken Kernels (%)
Bidì 03	77.43 ± 1.80 a	58.57 ± 0.74 bc	21.00 ± 1.41 fgh	0.00 ± 0.00	0.00 ± 0.00 d
Simeto	75.17 ± 0.35 ab	66.63 ± 0.99 a	6.00 ± 2.83 h	0.50 ± 0.71	7.00 ± 0.00 a–d
Bivona 03	72.93 ± 0.71 abc	53.40 ± 0.98 c–h	72.50 ± 3.54 ab	0.00 ± 0.00	2.50 ± 2.12 d
Bivona 04	74.10 ± 1.13 ab	54.73 ± 0.58 c–g	58.00 ± 1.41 a–d	0.50 ± 0.71	8.50 ± 2.12 a–d
Bufala Bianca 02	73.20 ± 0.61 abc	50.97 ± 2.29 e–h	68.00 ± 2.83 abc	0.50 ± 0.71	2.00 ± 1.41 d
Bufala Bianca 03	71.90 ± 0.95 abc	51.63 ± 2.29 d–h	40.00 ± 2.83 d–g	0.50 ± 0.71	6.50 ± 2.12 a–d
Bufala Nera Corta 01	71.30 ± 1.65 bc	55.13 ± 1.05 c–f	34.00 ± 5.66 d–g	2.00 ± 1.41	14.50 ± 0.71 a
Bufala Nera Corta 02	71.43 ± 1.52 bc	56.93 ± 1.89 bcd	44.00 ± 2.83 c–f	1.50 ± 0.71	14.00 ± 1.41 a
Bufala Nera Lunga 01	71.33 ± 1.99 bc	55.87 ± 1.75 b–e	37.50 ± 2.12 d–g	0.00 ± 0.00	9.00 ± 1.41 a–d
Bufala Nera Lunga 02	74.43 ± 0.50 ab	58.93 ± 0.59 bc	32.50 ± 3.54 efg	0.50 ± 0.71	3.00 ± 1.41 cd
Bufala Nera Lunga 04	73.43 ± 0.75 abc	55.93 ± 1.76 b–e	47.00 ± 5.66 cde	0.00 ± 0.00	2.00 ± 1.41 d
Bufala Rossa Corta b01	72.97 ± 1.66 abc	51.60 ± 2.72 d–h	21.00 ± 0.00 fgh	0.50 ± 0.71	12.50 ± 2.12 abc
Bufala Rossa Lunga 01	68.23 ± 1.26 c	49.53 ± 0.59 fgh	18.00 ± 2.83 gh	0.00 ± 0.00	13.00 ± 2.83 ab
Bufala Rossa Lunga 03	75.03 ± 2.64 ab	52.47 ± 1.10 d–h	33.50 ± 0.71 d–g	1.00 ± 0.00	4.00 ± 1.41 bcd
Bufala Cerami 01	75.93 ± 0.47 ab	47.80 ± 0.44 h	49.50 ± 2.12 b–e	0.00 ± 0.00	4.00 ± 1.41 bcd
Bufala Salice 01	73.03 ± 1.62 abc	61.60 ± 0.62 ab	54.50 ± 3.54 a–e	1.50 ± 0.71	2.50 ± 2.12 d
Bufala Troina 01	74.20 ± 1.70 ab	54.23 ± 0.23 c–g	49.00 ± 7.07 b–e	1.00 ± 0.00	5.00 ± 1.41 a–d
Ciciredda 02	71.20 ± 0.95 bc	49.23 ± 1.06 gh	40.00 ± 1.41 d–g	0.00 ± 0.00	13.00 ± 1.41 ab
Ciciredda 03	73.07 ± 0.64 abc	53.20 ± 1.04 c–h	77.00 ± 1.41 a	0.50 ± 0.71	6.00 ± 1.41 a–d
Paola 02	70.77 ± 0.99 bc	56.53 ± 1.34 b–e	75.00 ± 11.31 a	0.50 ± 0.71	2.00 ± 1.41 d

**Table 2 plants-12-02641-t002:** Physico-chemical characteristics of the semolina in the studied genotypes (data are means ± standard deviations). For all the response variables, different letters indicate a significant difference between the means (*p* ≤ 0.001).

Sample	Moisture (g/100 g)	Protein (g/100 g d.m.)	Ash (g/100 g d.m.)	Wet Gluten (g/100 g)	Dry Gluten (g/100 g)	Gluten Index	Water Binding in Wet Gluten (g/100 g)
Bidì 03	14.30 ± 0.14	12.53 ± 0.03 cd	1.04 ± 0.01 abc	27.45 ± 0.21 a–d	9.15 ± 0.07 a–d	80.34 ± 3.97 ab	18.30 ± 0.14 a–d
Simeto	15.00 ± 0.15	13.06 ± 0.02 bc	0.99 ± 0.01 a–e	26.57 ± 0.32 a–d	9.18 ± 0.06 a–d	95.22 ± 1.80 ab	17.39 ± 0.25 a–d
Bivona 03	15.10 ± 0.42	10.96 ± 0.13 ef	0.86 ± 0.02 gh	21.54 ± 0.34 cd	6.67 ± 0.21 e	73.23 ± 0.15 ab	14.86 ± 0.13 a–d
Bivona 04	15.30 ± 0.35	12.01 ± 0.01 d	0.85 ± 0.00 h	23.78 ± 1.83 a–d	7.51 ± 0.55 cde	80.65 ± 0.91 ab	16.27 ± 1.28 a–d
Bufala Bianca 02	15.20 ± 0.42	11.72 ± 0.13 de	0.87 ± 0.01 f–h	19.89 ± 0.08 d	6.22 ± 0.05 e	65.06 ± 4.32 abc	13.67 ± 0.03 d
Bufala Bianca 03	14.70 ± 0.42	10.98 ± 0.09 ef	0.96 ± 0.01 b–h	25.59 ± 0.83 a–d	8.14 ± 0.28 a–e	65.24 ± 7.63 abc	17.46 ± 0.55 a–d
Bufala Nera Corta 01	14.90 ± 0.71	14.43 ± 0.10 a	0.96 ± 0.01 b–h	22.89 ± 1.10 bcd	7.74 ± 0.22 b–e	28.38 ± 10.18 def	15.15 ± 0.88 a–d
Bufala Nera Corta 02	15.35 ± 0.35	12.08 ± 0.03 cd	0.88 ± 0.00 e–h	24.59 ± 0.41 a–d	8.21 ± 0.15 a–e	51.96 ± 7.61 b–e	16.38 ± 0.26 a–d
Bufala Nera Lunga 01	15.10 ± 0.15	14.06 ± 0.09 a	0.96 ± 0.02 b–h	28.63 ± 0.36 abc	9.40 ± 0.30 a–d	52.51 ± 7.26 b–e	19.23 ± 0.06 a–d
Bufala Nera Lunga 02	15.20 ± 0.14	13.89 ± 0.06 ab	0.97 ± 0.01 a–g	31.21 ± 2.25 a	10.23 ± 0.61 a	55.96 ± 8.57 bcd	20.98 ± 1.63 ab
Bufala Nera Lunga 04 (*)	15.10 ± 0.29	9.77 ± 0.38 g	0.97 ± 0.03 a–g	28.64 ± 3.09 abc	7.55 ± 0.00 cde	23.40 ± 2.33 ef	21.09 ± 3.08 a
Bufala Rossa Corta b01	15.20 ± 0.28	14.24 ± 0.34 a	1.00 ± 0.02 a–d	30.59 ± 0.29 ab	9.93 ± 0.26 ab	54.49 ± 2.24 b–e	20.66 ± 0.03 abc
Bufala Rossa Lunga 01	15.50 ± 0.71	13.68 ± 0.04 ab	1.08 ± 0.02 a	21.94 ± 2.45 cd	7.38 ± 0.92 de	33.16 ± 0.62 def	14.56 ± 1.54 a–d
Bufala Rossa Lunga 03	15.40 ± 0.28	14.34 ± 0.06 a	0.93 ± 0.01 c–h	28.58 ± 2.59 abc	9.72 ± 0.43 abc	35.55 ± 5.02 c–f	18.86 ± 2.16 a–d
Bufala Cerami 01 (*)	15.20 ± 0.28	10.68 ± 0.01 fg	0.98 ± 0.01 a–f	19.63 ± 1.14 d	6.36 ± 0.45 e	17.91 ± 6.56 f	13.27 ± 0.70 d
Bufala Salice 01 (*)	15.10 ± 0.42	14.60 ± 0.02 a	0.95 ± 0.02 b–h	21.80 ± 0.87 cd	7.28 ± 0.42 de	37.25 ± 2.36 c–f	14.55 ± 0.45 bcd
Bufala Troina 01 (*)	15.20 ± 0.57	10.98 ± 0.18 ef	0.95 ± 0.00 b–h	22.80 ± 0.92 bcd	7.50 ± 0.46 cde	28.06 ± 0.60 def	15.26 ± 0.46 a–d
Ciciredda 02	15.20 ± 0.71	13.06 ± 0.33 bc	1.05 ± 0.00 ab	23.54 ± 0.99 a–d	7.80 ± 0.20 b–e	28.88 ± 9.08 def	15.74 ± 0.79 a–d
Ciciredda 03	15.20 ± 0.42	9.70 ± 0.28 g	0.93 ± 0.02 d–h	20.90 ± 0.17 cd	6.49 ± 0.14 e	52.48 ± 0.38 b–e	14.42 ± 0.31 cd
Paola 02	15.50 ± 0.42	11.94 ± 0.03 de	0.91 ± 0.01 d–h	25.39 ± 0.64 a–d	7.85 ± 0.29 b–e	39.22 ± 0.61 c–f	17.54 ± 0.35 a–d

(*) Indicates samples for which the gluten was extracted manually, centrifuged with Centrifuge 2015 and dried with Glutork 2020.

**Table 3 plants-12-02641-t003:** Colorimetric indices of the semolina in the studied genotypes (data are means ± standard deviations). For all the response variables, different letters indicate a significant difference between the means (*p* ≤ 0.001).

Sample	Luminosity (*L**)	Red Index (*a**)	Yellow Index (*b**)
Bidì 03	89.58 ± 0.01 j	−7.38 ± 0.01 h	21.12 ± 0.02 c
Simeto	88.68 ± 0.01 k	−7.62 ± 0.01 i	23.21 ± 0.01 a
Bivona 03	91.40 ± 0.00 d	−7.07 ± 0.02 f	16.23 ± 0.01 p
Bivona 04	91.42 ± 0.05 d	−7.36 ± 0.02 h	17.78 ± 0.02 h
Bufala Bianca 02	91.30 ± 0.00 de	−7.23 ± 0.02 g	17.44 ± 0.01 j
Bufala Bianca 03	89.58 ± 0.00 j	−6.76 ± 0.01 b	18.02 ± 0.03 e
Bufala Nera Corta 01	91.00 ± 0.01 g	−6.98 ± 0.01 de	17.15 ± 0.01 l
Bufala Nera Corta 02	91.59 ± 0.01 c	−7.04 ± 0.02 ef	16.17 ± 0.02 p
Bufala Nera Lunga 01	89.66 ± 0.01 j	−6.74 ± 0.02 b	17.87 ± 0.01 g
Bufala Nera Lunga 02	89.90 ± 0.01 i	−7.19 ± 0.01 g	19.10 ± 0.01 d
Bufala Nera Lunga 04	91.19 ± 0.03 ef	−7.16 ± 0.01 g	17.51 ± 0.01 i
Bufala Rossa Corta b01	88.82 ± 0.01 k	−6.92 ± 0.02 cd	21.45 ± 0.02 b
Bufala Rossa Lunga 01	89.86 ± 0.01 i	−6.59 ± 0.01 a	17.75 ± 0.02 h
Bufala Rossa Lunga 03	91.44 ± 0.01 cd	−7.32 ± 0.00 h	17.95 ± 0.02 f
Bufale Cerami 01	91.22 ± 0.01 ef	−7.08 ± 0.00 f	17.36 ± 0.01 k
Bufale Salice 01	91.07 ± 0.01 fg	−6.84 ± 0.01 c	16.23 ± 0.01 p
Bufale Troina 01	90.80 ± 0.01 b	−7.22 ± 0.04 g	16.78 ± 0.01 m
Ciciredda 02	90.53 ± 0.01 h	−6.59 ± 0.01 a	16.39 ± 0.02 o
Ciciredda 03	92.25 ± 0.01 a	−7.17 ± 0.04 g	16.03 ± 0.02 q
Paola 02	91.33 ± 0.01 de	−7.01 ± 0.03 ef	16.56 ± 0.04 n

**Table 4 plants-12-02641-t004:** Technological quality characteristics of the semolina in the studied genotypes (data are means ± standard deviations). For all the response variables, different letters indicate a significant difference between the means (*p* ≤ 0.001).

			Mixograph	
Sample	SDS Sedimentation Height (mm)	Falling Number (s)	Mixing Time (s)	Peak Dough Height (M.U.)
Bidì 03	32.00 ± 1.41 abc	457.00 ± 1.41 d	127.00 ± 7.07 g	4.90 ± 0.14 a
Simeto	35.50 ± 0.71 a	460.00 ± 0.00 d	236.00 ± 0.00 b	4.50 ± 0.14 ab
Bivona 03	25.50 ± 0.71 f–h	482.50 ± 0.71 c	207.50 ± 10.61 bcd	3.40 ± 0.14 cde
Bivona 04	29.50 ± 0.71 b–f	458.50 ± 0.71 d	209.00 ± 4.24 bcd	3.60 ± 0.28 cde
Bufala Bianca 02	27.00 ± 0.00 d–g	421.50 ± 2.12 i	236.00 ± 8.49 b	3.20 ± 0.14 de
Bufala Bianca 03	30.00 ± 0.00 b–e	409.50 ± 2.12 j	175.50 ± 2.12 c–g	3.45 ± 0.07 cde
Bufala Nera Corta 01	31.50 ± 0.71 abc	443.00 ± 1.41 ef	162.00 ± 2.83 d–g	3.15 ± 0.21 de
Bufala Nera Corta 02	28.00 ± 1.41 c–g	370.00 ± 1.41 k	190.00 ± 14.14 b–e	3.35 ± 0.07 cde
Bufala Nera Lunga 01	30.00 ± 0.00 b–e	446.00 ± 0.00 e	195.00 ± 14.14 b–e	3.80 ± 0.14 b–e
Bufala Nera Lunga 02	29.50 ± 0.71 b–f	491.50 ± 0.71 b	147.50 ± 7.78 efg	3.65 ± 0.07 b–e
Bufala Nera Lunga 04	26.00 ± 0.00 e–h	436.00 ± 1.41 fg	176.50 ± 2.12 c–f	2.95 ± 0.07 e
Bufala Rossa Corta b01	31.00 ± 0.00 bcd	509.00 ± 1.41 a	157.50 ± 3.54 efg	3.85 ± 0.07 bcd
Bufala Rossa Lunga 01	33.00 ± 1.41 ab	420.50 ± 0.71 i	210.00 ± 14.14 bcd	3.35 ± 0.21 cde
Bufala Rossa Lunga 03	26.00 ± 0.00 e–h	309.50 ± 2.12 m	138.50 ± 2.12 fg	4.10 ± 0.14 abc
Bufala Cerami 01	24.00 ± 0.00 gh	431.00 ± 1.41 gh	169.00 ± 7.07 c–g	3.80 ± 0.14 b–e
Bufala Salice 01	30.00 ± 0.00 b–e	455.00 ± 1.41 d	290.00 ± 14.14 a	3.65 ± 0.21 b–e
Bufala Troina 01	22.00 ± 0.00 h	412.00 ± 1.41 j	189.50 ± 7.78 b–e	3.10 ± 0.14 de
Ciciredda 02	30.50 ± 0.71 bcd	432.50 ± 0.71 g	175.00 ± 0.00 c–g	3.50 ± 0.00 cde
Ciciredda 03	25.50 ± 0.71 f–h	349.00 ± 1.41 l	215.00 ± 7.07 bc	3.35 ± 0.07 cde
Paola 02	25.50 ± 0.71 f–h	424.50 ± 0.71 hi	165.50 ± 3.54 d–g	3.65 ± 0.07 b–e

**Table 5 plants-12-02641-t005:** Alveograph and farinograph indices of the doughs in the studied genotypes (data are means ± standard deviations). For all the response variables, different letters indicate a significant difference between the means (*p* ≤ 0.001).

Sample	Alveograph		Farinograph			
	W (10^−4^ × J)	P/L	H_2_O Absorption in 500 B.U. (g/100 g)	Development Time (min)	Dough Stability (min)	Degree of Softening (B.U.)
Bidì 03	87.50 ± 7.78 b	1.34 ± 0.06 bc	60.60 ± 0.14 b	2.00 ± 0.14 ab	1.75 ± 0.07 cd	95.00 ± 2.83 i
Simeto	145.00 ± 4.24 a	3.93 ± 0.25 a	63.20 ± 0.14 a	2.20 ± 0.14 a	4.60 ± 0.14 a	53.00 ± 2.83 j
Bivona 03	42.00 ± 4.24 cde	0.62 ± 0.18 ef	51.25 ± 0.07 k	1.30 ± 0.00 de	1.20 ± 0.14 d–g	122.00 ± 4.24 fgh
Bivona 04	48.00 ± 2.83 c	0.77 ± 0.08 c–f	53.10 ± 0.00 g	1.50 ± 0.00 c–e	1.55 ± 0.07 c–f	97.00 ± 1.41 hi
Bufala Bianca 02	22.50 ± 3.54 def	0.58 ± 0.06 ef	49.00 ± 0.00 m	1.40 ± 0.00 c–e	1.40 ± 0.00 d–g	123.00 ± 4.24 fgh
Bufala Bianca 03	35.00 ± 4.24 c–f	0.44 ± 0.06 f	52.35 ± 0.07 h	1.70 ± 0.14 bc	2.40 ± 0.14 b	101.00 ± 5.66 ghi
Bufala Nera Corta 01	19.00 ± 4.24 ef	1.22 ± 0.16 b–e	54.60 ± 0.00 f	1.45 ± 0.07 c–e	0.85 ± 0.07 g	201.00 ± 4.24 a
Bufala Nera Corta 02	25.00 ± 4.24 c–f	1.30 ± 0.14 bcd	52.40 ± 0.00 h	1.20 ± 0.00 e	1.30 ± 0.14 d–g	133.00 ± 4.24 def
Bufala Nera Lunga 01	35.00 ± 5.66 c–f	0.98 ± 0.04 c–f	54.60 ± 0.00 f	1.70 ± 0.00 bc	1.20 ± 0.14 d–g	183.00 ± 2.83 ab
Bufala Nera Lunga 02	46.50 ± 3.54 cd	0.88 ± 0.04 c–f	57.75 ± 0.07 e	1.70 ± 0.00 bc	1.35 ± 0.07 d–g	157.00 ± 5.66 bcd
Bufala Nera Lunga 04	12.00 ± 4.24 f	0.97 ± 0.04 c–f	48.80 ± 0.00 m	1.30 ± 0.00 de	1.05 ± 0.07 fg	138.00 ± 4.24 def
Bufala Rossa Corta b 01	73.50 ± 3.54 b	1.34 ± 0.06 bc	59.90 ± 0.00 c	1.90 ± 0.00 ab	1.50 ± 0.00 c–f	120.00 ± 1.41 fgh
Bufala Rossa Lunga 01	40.00 ± 2.83 cde	1.68 ± 0.06 b	58.35 ± 0.07 d	1.50 ± 0.00 c–e	1.45 ± 0.07 c–f	140.00 ± 5.66 def
Bufala Rossa Lunga 03	31.00 ± 2.83 c–f	0.72 ± 0.04 c–f	52.30 ± 0.00 h	1.65 ± 0.07 bcd	1.30 ± 0.00 d–g	172.00 ± 2.83 bc
Bufala Cerami 01	20.00 ± 2.83 ef	0.97 ± 0.18 c–f	51.55 ± 0.07 ijk	1.65 ± 0.07 bcd	1.35 ± 0.07 d–g	150.00 ± 7.07 cde
Bufala Salice 01	31.00 ± 4.24 c–f	0.85 ± 0.10 c–f	51.75 ± 0.07 i	1.45 ± 0.07 c–e	1.65 ± 0.07 cde	133.00 ± 4.24 def
Bufala Troina 01	20.50 ± 4.95 ef	0.66 ± 0.09 def	50.75 ± 0.07 l	1.50 ± 0.00 c–e	1.55 ± 0.07 c–f	146.00 ± 5.66 c–f
Ciciredda 02	37.00 ± 2.83 c–f	0.98 ± 0.04 c–f	54.75 ± 0.07 f	1.50 ± 0.00 c–e	2.00 ± 0.14 bc	126.00 ± 4.24 efg
Ciciredda 03	32.50 ± 3.54 c–f	0.88 ± 0.11 c–f	51.65 ± 0.07 ij	1.40 ± 0.00 c–e	1.25 ± 0.07 d–g	134.00 ± 5.66 def
Paola 02	34.00 ± 4.24 c–f	0.62 ± 0.07 ef	51.30 ± 0.00 jk	1.20 ± 0.00 e	1.15 ± 0.07 e–g	142.00 ± 2.83 def

**Table 6 plants-12-02641-t006:** Quality characteristics of loaves in the studied genotypes (data are means ± standard deviations). For all the response variables, different letters indicate a significant difference between the means (*p* ≤ 0.001).

Sample	Volume (cm^3^)	Height (mm)	Weight (g)	Moisture (g/100 g)	Crumb Porosity *
Bidì 03	423.75 ± 1.77 a	73.50 ± 0.71 a	142.85 ± 0.00 b	30.12 ± 1.50	5.00 ± 0.00
Simeto	338.75 ± 1.77 g	65.50 ± 0.71 b–e	146.05 ± 0.00 a	34.04 ± 0.78	7.00 ± 0.00
Bivona 03	346.25 ± 1.77 fg	64.50 ± 0.71 c–f	135.10 ± 0.00 f	30.04 ± 0.13	7.00 ± 0.00
Bivona 04	397.50 ± 7.07 bc	70.50 ± 0.71 ab	138.20 ± 0.85 cd	32.31 ± 0.61	7.00 ± 0.00
Bufala Bianca 02	347.50 ± 3.54 fg	61.00 ± 1.41 d–g	134.38 ± 0.04 fg	28.40 ± 1.27	7.00 ± 0.00
Bufala Bianca 03	368.75 ± 1.77 de	59.50 ± 0.71 fgh	132.80 ± 0.00 g	31.44 ± 1.56	4.00 ± 0.00
Bufala Nera Corta 01	373.75 ± 1.77 d	59.50 ± 0.71 fgh	135.85 ± 0.00 ef	37.29 ± 9.93	7.00 ± 0.00
Bufala Nera Corta 02	362.50 ± 0.00 def	62.00 ± 0.00 d–g	138.08 ± 0.04 cd	30.85 ± 0.98	7.00 ± 0.00
Bufala Nera Lunga 01	393.75 ± 1.77 c	63.00 ± 0.00 def	139.50 ± 0.00 c	30.51 ± 0.72	5.00 ± 0.00
Bufala Nera Lunga 02	406.25 ± 1.77 abc	64.50 ± 0.71 c–f	142.35 ± 0.35 b	31.66 ± 2.09	6.00 ± 0.00
Bufala Nera Lunga 04	372.50 ± 3.54 de	56.50 ± 0.71 gh	134.90 ± 0.00 f	29.605 ± 0.39	7.00 ± 0.00
Bufala Rossa Corta b01	406.45 ± 1.48 abc	69.50 ± 0.71 abc	144.65 ± 0.00 a	32.36 ± 1.11	6.00 ± 0.00
Bufala Rossa Lunga 01	348.75 ± 1.77 fg	54.50 ± 0.71 h	141.80 ± 0.00 b	31.87 ± 0.89	6.00 ± 0.00
Bufala Rossa Lunga 03	415.00 ± 3.54 ab	70.00 ± 2.83 abc	139.45 ± 0.35 c	30.53 ± 0.03	6.00 ± 0.00
Bufale Cerami 01	368.75 ± 1.77 de	56.50 ± 0.71 gh	134.10 ± 0.00 fg	30.65 ± 0.42	6.00 ± 0.00
Bufale Salice 01	355.00 ± 3.54 efg	60.50 ± 0.71 efg	139.58 ± 0.04 c	32.81 ± 0.33	6.00 ± 0.00
Bufale Troina 01	393.75 ± 1.77 c	66.50 ± 0.71 bcd	135.15 ± 0.00 f	31.16 ± 0.17	5.00 ± 0.00
Ciciredda 02	346.25 ± 1.77 fg	60.50 ± 0.71 efg	137.10 ± 0.00 de	30.40 ± 0.60	7.00 ± 0.00
Ciciredda 03	360.00 ± 0.00 def	63.00 ± 0.00 def	135.15 ± 0.00 f	29.62 ± 0.97	6.00 ± 0.00
Paola 02	355.00 ± 7.07 efg	63.00 ± 1.41 def	137.98 ± 0.32 cd	30.60 ± 1.45	6.00 ± 0.00

* Scale 1–8; 1 = non-uniform structure, large and irregular cells; 8 = uniform compact structure, small and regular cells.

**Table 7 plants-12-02641-t007:** Colorimetric parameters of crust and crumb in the studied genotypes (data are means ± standard deviations) For all the response variables, different letters indicate a significant difference between the means (*p* ≤ 0.001).

Sample	Crumb			Crust		
	Luminosity (*L**)	Red Index (*a**)	Yellow Index (*b**)	Luminosity (*L**)	Red Index (*a**)	Yellow Index (*b**)
Bidì 03	71.25 ± 1.05 ab	−0.64 ± 0.04 c–f	18.91 ± 0.57 bcd	35.62 ± 1.41 bc	10.24 ± 2.27 bc	14.35 ± 1.33 d
Simeto	71.54 ± 0.13 ab	−0.12 ± 0.02 bc	22.97 ± 0.33 a	36.14 ± 1.85 bc	12.50 ± 0.69 abc	16.37 ± 1.84 cd
Bivona 03	75.84 ± 0.54 a	−1.90 ± 0.01 g	15.54 ± 0.06 f	36.59 ± 1.91 bc	12.94 ± 0.75 abc	17.00 ± 1.99 a–d
Bivona 04	74.15 ± 0.21 ab	−0.47 ± 0.16 c–f	18.24 ± 1.03 b–e	36.48 ± 0.41 bc	12.30 ± 0.33 abc	15.81 ± 0.58 d
Bufala Bianca 02	68.99 ± 2.16 ab	0.60 ± 0.01 a	17.67 ± 0.18 c–f	35.25 ± 2.28 c	12.41 ± 1.64 abc	15.47 ± 2.67 d
Bufala Bianca 03	67.61 ± 0.47 b	−0.21 ± 0.04 bcd	16.76 ± 0.61 def	36.05 ± 0.40 bc	13.12 ± 0.06 abc	16.49 ± 0.20 bcd
Bufala Nera Corta 01	71.28 ± 0.45 ab	−0.89 ± 0.18 ef	17.20 ± 0.18 c–f	38.60 ± 0.86 ab	15.69 ± 0.16 a	22.29 ± 0.91 a–d
Bufala Nera Corta 02	68.02 ± 0.39 b	1.08 ± 0.05 a	20.55 ± 0.18 ab	39.91 ± 0.01 ab	14.91 ± 0.01 ab	20.81 ± 0.78 a–d
Bufala Nera Lunga 01	71.94 ± 1.32 ab	−0.90 ± 0.12 ef	17.24 ± 0.32 c–f	43.15 ± 0.77 ab	15.65 ± 0.33 ab	24.60 ± 0.70 abc
Bufala Nera Lunga 02	72.08 ± 2.39 ab	−0.67 ± 0.23 c–f	17.29 ± 0.67 c–f	39.59 ± 1.51 ab	16.03 ± 1.04 a	22.45 ± 1.70 a–d
Bufala Nera Lunga 04	71.73 ± 0.40 ab	−1.84 ± 0.03 g	17.22 ± 0.14 c–f	39.47 ± 2.53 ab	9.05 ± 0.25 c	16.19 ± 1.51 cd
Bufala Rossa Corta b01	72.45 ± 2.09 ab	−0.95 ± 0.11 f	18.90 ± 0.41 bcd	35.88 ± 0.67 bc	12.60 ± 0.74 abc	15.07 ± 1.18 d
Bufala Rossa Lunga 01	67.49 ± 2.96 b	0.42 ± 0.08 ab	19.82 ± 0.69 bc	42.25 ± 1.89 ab	16.35 ± 0.14 a	25.47 ± 1.07 a
Bufala Rossa Lunga 03	69.68 ± 0.45 ab	−0.83 ± 0.21 def	17.42 ± 0.32 c–f	38.01 ± 1.07 ab	11.74 ± 0.69 abc	16.56 ± 0.08 bcd
Bufala Cerami 01	68.13 ± 0.07 b	−0.27 ± 0.15 cde	18.32 ± 0.12 b–e	42.67 ± 0.30 ab	15.12 ± 0.07 ab	24.49 ± 0.01 abc
Bufala Salice 01	72.72 ± 0.78 ab	−0.35 ± 0.04 c–f	17.70 ± 0.31 c–f	43.36 ± 0.36 ab	14.73 ± 0.40 ab	24.93 ± 0.71 ab
Bufala Troina 01	71.48 ± 0.10 ab	−2.17 ± 0.08 g	16.16 ± 0.01 ef	45.38 ± 0.18 a	13.29 ± 0.14 abc	21.55 ± 0.37 a–d
Ciciredda 02	73.00 ± 0.27 ab	−0.41 ± 0.13 c–f	16.32 ± 0.33 def	39.70 ± 0.50 ab	14.22 ± 0.08 abc	19.43 ± 0.59 a–d
Ciciredda 03	74.53 ± 0.57 ab	−1.83 ± 0.04 g	15.95 ± 0.13 ef	41.50 ± 1.00 ab	14.09 ± 2.03 abc	22.14 ± 3.35 a–d
Paola 02	72.55 ± 0.78 ab	−0.43 ± 0.04 c–f	17.42 ± 0.63 c–f	45.14 ± 1.69 a	13.26 ± 0.93 abc	21.75 ± 1.49 a–d

## Data Availability

All available data are reported in the paper.

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
