# Peer review of "Sicilian Rivet Wheat Landraces: Grain Characteristics and Technological Quality of Flour and Bread"

_plants, 2023, doi:10.3390/plants12142641_

Round 1
Reviewer 1 Report
The paper entitled "Commercial, physical-chemical and rheological features and bread-making behavior of Sicilian rivet wheat (Triticum turgidum subsp. turgidum)” reports numerous basic characteristics of 18 Sicilian durum wheat ecotypes and two durum wheat controls. The study should consider also a multivariate statistical analysis and provide innovative reflections of the results. The current approach leads to a sectorial research of limited interest for the general readers. The discussion considers limited comparisons with the literature and excludes interesting results formerly obtained in numerous studies on durum wheat. The English language should be revised.
Minor comments
Table 3, Table 7 and through all the text. Report L* (luminosity) instead of 100-L*.
Table 6. check the standard deviations of porosity.
Uniform the interline spaces of all the Tables.
Lines 312-324. Separate in a new paragraph, there are the Materials.
The English language should be revised.
Author Response
The paper entitled "Commercial, physical-chemical and rheological features and bread-making behavior of Sicilian rivet wheat (Triticum turgidum subsp. turgidum)” reports numerous basic characteristics of 18 Sicilian durum wheat ecotypes and two durum wheat controls. The study should consider also a multivariate statistical analysis and provide innovative reflections of the results. The current approach leads to a sectorial research of limited interest for the general readers. The discussion considers limited comparisons with the literature and excludes interesting results formerly obtained in numerous studies on durum wheat. The English language should be revised.
Thank you for revising our study so carefully. It helped us to improve the quality of the study.
As requested, we performed a multivariate statistical analysis (cluster analysis, in particular hierarchical cluster analysis and k-means), inserting an extensive comment and providing innovative reflections on the results in a new paragraph that we added (2.2). Furthermore, we added information on this particular multivariate analysis also in abstracts, key words, materials and methods, and conclusions.
To carry out this particular elaboration and the relative comment, we made use of the skills of our colleague Maria Allegra, particularly expert in statistics, of the other Research Centre always present in our headquarters in Acireale (Catania), Italy. Therefore, we have proposed it to you as co-author of any paper.
In addition, we have vastly improved the discussion and increased comparisons with the relevant literature.
As requested, we have revised the English language. In particular, Prof. Umberto Anastasi has revised the entire manuscript both as regards the contents and to check the English language. Therefore, we thought it appropriate to exchange the names of Paolo Guarnaccia (second co-author, instead of last) and Umberto Anastasi, last co-author, instead of second co-author of any paper.
Minor comments
Table 3, Table 7 and through all the text. Report L* (luminosity) instead of 100-L*.
As requested, both in Tables 3 and 7 as well as in the text, the term 100-L* was replaced with L* (luminosity). Also, the statistical analysis had been redone.
Table 6. check the standard deviations of porosity.
The standard deviation is equal to 0 because there is no difference between the values of the different replicates of each treatment. This due to the scale of Dallman used to make judgments to each replicates, which indicates the good quality of the analyses performed in the laboratory.
Uniform the interline spaces of all the Tables.
We arranged the format of the tables following your indication.
Lines 312-324. Separate in a new paragraph, there are the Materials
Now it’s adjusted.

Reviewer 2 Report
I carefully revised the manuscript concerning the characterization of eighteen Sicilian landraces of rivet wheat (Triticum turgidum subsp. turgidum). The focus of the research is contemporary, indeed wheat landraces represent an important source of traits suitable for sustainable agriculture. The paper contains several information about the Sicilian landraces, on the physical characteristics of grain (TKW, TW, kernel damage), on physicochemical parameters (moisture, proteins, ash, gluten index, gluten content) and on rheological characteristics and bread-making aptitude. Interesting findings on the characteristics of these landraces, useful to drive their end-use, are present. However, the way the results are presented makes the article unattractive. Indeed, the content is only observatory than explanatory and the discussion of the results is insufficient. I suggest the Authors to present the results in more attractive way, not only “higher value was observed in….lower value was observed in…”. The paper seems to be prepared careless and mistakes are present.
An analysis to discriminate or to group the landraces might be useful to show the results.
Minor suggestions:
Title: use the term physico-chemical
Line 128: Alternaria alternata
Table 1: title Physical characteristics of kernel in the Sicilian landraces
Line 155: when the authors cite legal limits, they have to insert the values
Line 201: yellow index is a commercial and not nutritional trait
Lines 199-202: rephrase the sentence because Simeto and Bufala Rossa Corta b01 have the higher b* values
Table 3: why the term semolina is used? It is wholewheat flour
Line 208 and others: often the authors cite themselves in the text. It’s better to avoid, it is unnecessary.
Line 250-259/285-302: Results should be reported in full and discussed.
Line 283: colourimetric data of the breads
Paragraph 3.1 insert the materials in a separate paragraph from methods
Line 336: Cyclotec type 120 (Foss)
Paragraph 3.2.1 Use the term moisture instead of humidity
English have to be completely revised
Author Response
I carefully revised the manuscript concerning the characterization of eighteen Sicilian landraces of rivet wheat (Triticum turgidum subsp. turgidum). The focus of the research is contemporary, indeed wheat landraces represent an important source of traits suitable for sustainable agriculture. The paper contains several information about the Sicilian landraces, on the physical characteristics of grain (TKW, TW, kernel damage), on physicochemical parameters (moisture, proteins, ash, gluten index, gluten content) and on rheological characteristics and bread-making aptitude. Interesting findings on the characteristics of these landraces, useful to drive their end-use, are present. However, the way the results are presented makes the article unattractive. Indeed, the content is only observatory than explanatory and the discussion of the results is insufficient. I suggest the Authors to present the results in more attractive way, not only “higher value was observed in….lower value was observed in…”. The paper seems to be prepared careless and mistakes are present.
Thanks for your precious revision work, and the useful suggestions.
As requested by the reviewer, we have made the manuscript more attractive as we have improved and expanded the discussion. We have also removed all errors still present on the manuscript.
As requested, we performed a new statistical analysis (cluster analysis, in particular hierarchical cluster analysis and k-means), inserting an extensive comment and providing innovative reflections on the results in a new paragraph that we added (2.2). Furthermore, we added information on this particular multivariate analysis also in abstracts, key words, materials and methods, and conclusions.
To carry out this particular elaboration and the relative comment, we made use of the skills of our colleague Maria Allegra, particularly expert in statistics, of the other Research Centrer always present in our headquarters in Acireale (Catania), Italy. Therefore, we have proposed it to you as co-author of any paper.
In addition, we have vastly improved all sections of the manuscript. In particular, Prof. Umberto Anastasi has revised the entire manuscript as regards the contents. Therefore, we thought it appropriate to exchange the names of Paolo Guarnaccia (second co-author, instead of last) and Umberto Anastasi, last co-author, instead of second co-author of any paper.
Minor suggestions:
Title: use the term physico-chemical
Thank you. We used this term, both in the title and key words.
Line 128: Alternaria alternata
Done, thank you for the suggestion.
Table 1: title Physical characteristics of kernel in the Sicilian landraces
Thank you. We changed the title of the table, as you suggested.
Line 155: when the authors cite legal limits, they have to insert the values
As requested, we inserted the range value (from 14.50% to 15.50 % for moisture; 10.50 for protein; 0.90 for ash) established by the Italian law (DPR 187/2001). Thank you for the advice.
Line 201: yellow index is a commercial and not a nutritional trait
Thank you so much for this insight. We changed the sentence, to make clear that although the yellow index is primarily a commercial trait, it also gives a piece of information about the nutraceutical value of the product, since the higher the yellow index, the higher the content of carotenoids, which are important bioactive compounds with beneficial effects on health.
Lines 199-202: rephrase the sentence because Simeto and Bufala Rossa Corta b01 have the higher b* values
Sorry, we correctly rephrased the sentence.
Table 3: why the term semolina is used? It is wholewheat flour
We thank the reviewer for his observation thanks to which we realized that almost always in the text we did not use the correct semolina terminology. Therefore, we have left the term semolina in table 3 and corrected the incorrect term (whole flour) with the correct term semolina throughout the text of the manuscript.
Line 208 and others: often the authors cite themselves in the text. It’s better to avoid, it is unnecessary.
As suggested, we deleted the following references: no 38 Spina et al., (2017); no 42-43. Spina et al., (2001). n° 59; Canale et al., (2022). The other our citations in the text are necessary in terms of comparison with studies carried out on the same topics of this paper.
Line 250-259/285-302: Results should be reported in full and discussed.
Thank you for the suggestion. We discussed the results of the farinograph analysis and bread colour more completely.
Line 283: colourimetric data of the breads
Thank you, now it is adjusted.
Paragraph 3.1 insert the materials in a separate paragraph from methods
Thank you, now it is adjusted.
Line 336: Cyclotec type 120 (Foss)
We thank the reviewer for this observation which allowed us to correct a typo regarding the type of experimental mill used. To obtain the semolina we used the experimental mill 'Bona' (Monza, Italy).
Paragraph 3.2.1 Use the term moisture instead of humidity
Thank you, now it is adjusted.

Round 2
Reviewer 1 Report
Even if the Cluster Analysis was included, the discussion is still poor for the lack of comparisons with the literature.
In fact, the Authors did not improve the paper considering the following suggestion:
The discussion considers limited comparisons with the literature and excludes interesting results formerly obtained in numerous studies on durum wheat.
For example, several papers formerly studied the cv. Simeto.
Line 361. What is “A tandem of Cluster analysis”
Colour parameters L*, a*, b* must be in italics.
Some Tables of the Cluster analysis should be shown as Supplementary materials.
The English language still needs to be improved. For example, avoid direct statements such as “The results from such analyses allowed the authors to evaluate the genotypes bread-making suitability” (lines 34-359). Use impersonal (third person).
Author Response
Even if the Cluster Analysis was included, the discussion is still poor for the lack of comparisons with
the literature.
In fact, the Authors did not improve the paper considering the following suggestion:
The discussion considers limited comparisons with the literature and excludes interesting results
formerly obtained in numerous studies on durum wheat.
For example, several papers formerly studied the cv. Simeto.
Thank you. As requested, we have improved the manuscript, especially in the Results and Discussion
section. In particular, we included several citations and compared our results with those reported in
the cited literature. Since we compared the genotypes of the turgidum subspecies with the control
varieties ‘Bidì’ and ‘Simeto’ of the durum subspecies, we have used as references the papers that
studied these last genotypes.
As requested, we carried out further English editing. During this English editing process, we thought
of editing the title as well: ‘Sicilian rivet wheat landraces: grain characteristics and technological
quality of flour and bread’.
However, considering that the reviewers were fine with the first version of the title, we can leave the
first version, for us it is indifferent. We leave the final choice to you.
Line 361. What is “A tandem of Cluster analysis”
A tandem of Cluster analysis consists of more than one statistical analysis. In our case, they are two
statistical analyses (Hierarchical Cluster analysis and k-means Cluster analysis) used in tandem, e.g.,
in sequence.
Colour parameters L*, a*, b* must be in italics.
Thank you, it was corrected.
Some Tables of the Cluster analysis should be shown as Supplementary materials.
Thank you. As requested, we have moved Tables 8 and 9 into Supplementary Materials and corrected
the name of these tables in the text.

Reviewer 2 Report
Authors improved the Manuscript as suggested
Author Response
Authors improved the Manuscript as suggested.
We thank the reviewer very much for appreciating the work we did on the manuscript and for saying that the manuscript was fine.

Round 3
Reviewer 1 Report
I prefer the second title.
The English language was improved